# The longitudinal association between patient empowerment and patient-reported outcomes: What is the direction of effect?

**Mariela Acuña Mora**[1,2], **Carina Sparud-Lundin**[1], **Eva Fernlund**[3,4], **Shalan Fadl**[5], **Kazamia Kalliopi**[6], **Annika Rydberg**[7], **Åsa Burström**[8], **Katarina Hanseus**[9], **Philip Moons**[1,10,11], **Ewa-Lena Bratt**[1,12]*

1 Institute of Health and Care Sciences, University of Gothenburg, Gothenburg, Sweden, 2 Faculty of Caring Science, Work Life and Social Welfare, University of Borås, Borås, Sweden, 3 Division of Pediatrics, Department of Clinical and Experimental Medicine, Linköping University, Linköping, Sweden, 4 Crown Princess Victoria Children's Hospital, Linköping University Hospital, Linköping, Sweden, 5 Department of Children and Young Adults, University Hospital Örebro, Örebro, Sweden, 6 Department of Women's and Children's Health, Karolinska Institute, Stockholm, Sweden, 7 Department of Clinical Sciences, Umeå University, Umeå, Sweden, 8 Karolinska Institute, Department of Neurobiology, Care Sciences and Society, Stockholm, Sweden, 9 Children's Heart Center, Skåne University Hospital Lund, Lund, Sweden, 10 Department of Public Health and Primary Care, KU Leuven, Leuven, Belgium, 11 Department of Paediatrics and Child Health, University of Cape Town, Cape Town, South Africa, 12 Department of Pediatric Cardiology, The Queen Silvia Children's Hospital, Gothenburg, Sweden

* ewa-lena.bratt@gu.se

## Abstract

### Background

Theoretical literature and cross-sectional studies suggest empowerment is associated with other patient-reported outcomes (PROs). However, it is not known if patient empowerment is leading to improvements in other PROs or vice versa.

### Aims

The present study aimed to examine the direction of effects between patient empowerment and PROs in young persons with congenital heart disease (CHD).

### Methods

As part of the STEPSTONES-CHD trial, adolescents with CHD from seven pediatric cardiology centers in Sweden were included in a longitudinal observational study (n = 132). Data were collected when patients were 16 (T0), 17 (T1) and 18 ½ years old (T2). The Gothenburg Young Persons Empowerment Scale (GYPES) was used to measure patient empowerment. Random intercepts cross-lagged panel models between patient empowerment and PROs (communication skills; patient-reported health; quality of life; and transition readiness) were undertaken.

**Data Availability Statement:** Data cannot be shared publicly because of the sensitive nature of it. However, requests to access the data set from

qualified researchers trained in human subject confidentiality protocols can be made by sending an email to the University of Gothenburg at karin. dejke@gu.se.

**Funding:** This study is supported by research grants from the Swedish Research Council for Health, Working Life and Welfare-FORTE (grant STYA-2015/0003; http://forte.se/; PM); Swedish Heart-Lung Foundation (grant 20150535; https:// www.hjart-lungfonden.se/; PM); Swedish Research Council (grant 2015-02503; https://www.vr.se/; PM); Swedish Children Heart Association (http:// www.hjartebarn.se/; ELB). The funders had no role in study design, data collection and analysis, decision to publish, or preparation of the manuscript.

**Competing interests:** The authors have declared that no competing interests exist.

## Results

We found a significant cross-lagged effect of transition readiness over patient empowerment between T1 and T2, signifying that a higher level of transition readiness predicted a higher level of patient empowerment. No other significant cross-lagged relationships were found.

## Conclusion

Feeling confident before the transition to adult care is necessary before young persons with CHD can feel in control to manage their health and their lives. Clinicians interested in improving patient empowerment during the transitional period should consider targeting transition readiness.

## Background

An increase in the prevalence of chronic conditions (CC) has led healthcare systems to develop care models that accommodate the treatment of patients who are in need of follow-up, multi-disciplinary attention and in some cases frequent medical interventions [1,2]. In recent years, person-centered care has been suggested and introduced. This entails healthcare providers collaborating with the patients, planning mutual care goals and involving them in the decision-making process [3], aspects that are important when treating those with CC. Nevertheless, those who live with CC also need to become autonomous, learn to cope with their CC and develop critical skills to manage their condition [4,5].

Patient empowerment has been suggested as a way of achieving the aforementioned goals [6]. This construct can be understood as a process or outcome that results from communicating with the healthcare professional, which increases the patient's sense of control, coping abilities and self-efficacy [7]. Empowerment has been theoretically and empirically associated with improvements on other patient-reported and clinical outcomes [8]. It entails the person becoming the manager of their own care, having the skills to set goals and define ways of achieving them and gaining higher knowledge, which is expected to lead them to making healthier choices [9]. This makes it a relevant outcome for those with a CC. Moreover, patient empowerment is a central concept within nursing science, given that nurses are responsible for helping individuals feel capable of caring for themselves based on their life's resources [10].

Studies that investigated the association between patient empowerment and clinical outcomes have found that a higher level of patient empowerment is associated with improved glycemic control (e.g. HbA1c, glucose testing, foot care) [11–16] and better disease and pain management [13]. Additionally, studies have also found that patient empowerment is associated with patient-reported outcomes (PROs), including disease-related knowledge [17,18], self-efficacy [19,20], health behaviors [11], transition readiness [21], communication skills [21], self-care [11] and quality of life (QoL) [22–26]. The problem with these studies is that we do not know the direction of effect. For instance, is more knowledge about the condition resulting in a higher level of patient empowerment, or a higher empowerment level triggering people to seek information and consequently build up more knowledge? Such direction of effect can only be investigated by using longitudinal designs. Understanding directionality can clarify how different variables are associated with each other and which one has a predictive effect. This information is important when developing interventions and determining which outcomes should be targeted and possible effects over other variables.

A particular group that can benefit from patient empowerment, is young persons with CC. During adolescence this group will face important physical and psychosocial changes. They will also have to fulfill tasks associated with this developmental stage, as well as prepare for adulthood, all while dealing with the effects of their CC. Increasing their level of empowerment can help this group develop essential skills that will facilitate these changes and tasks. Congenital heart defects (CHD) are the most frequent congenital disorders [27] and represent a heterogeneous group, in terms of defects, complexity, treatment and long-term needs [28]. Patients with CHD therefore serve as a good sample case when evaluating the generalizability of results. The present study thus aimed to examine the longitudinal associations between patient empowerment and PROs in young persons with CHD.

## Methods

### Design

This study is part of a larger study evaluating the effectiveness of a person-centered transition program to empower young persons with CHD using a hybrid experimental design [29]. This design involves a randomized controlled trial, where patients are assigned to an intervention or comparison group and embedded in a longitudinal, observational study, which comprises a control group from intervention-naive centers. The study evaluates the effectiveness of a nurse-led intervention who received training amongst other things on adolescent health, person-centered care, and sexual and reproductive health. Participants were followed up for a period of two and a half years and were asked to answer a set of questionnaires at three different time points. Given the aim of the present study, only the comparison and control groups were included, as the patients in these groups were not exposed to any intervention that could increase their level of empowerment. Additional information on the hybrid experimental design can be found in a methods paper [29].

### Study population

Eligible participants had to fulfill the following criteria: 1) have been diagnosed with a CHD; 2) age 16 years; 3) Swedish speaking; and 4) literate [29]. Young persons were excluded if they had cognitive and/or physical limitations inhibiting them from answering the questionnaires or had a prior heart transplantation.

Given that this study is part of a larger project, the sample size was calculated based on the primary outcome of the larger study, which was patient empowerment. The target was an improvement of 5.25 points (i.e., half a standard deviation). For two-sided tests with $\alpha = 0.05$ and power = 80%, 63 patients were needed in each arm of the RCT. To compensate for potential drop-out, 70 were recruited in each arm [29]. Combining the patients from the comparison arm (n = 70) and the control arm (n = 70) together, the potential total sample was of 140 participants [29]. A total of 138 individuals answered the questionnaires at T0, 108 at T1 and 101 at T2. Between T0 and T1, 29 participants dropped-out of the study and between T1 and T2 an additional seven patients. However, these were still included in the analyses to increase the sample size. Drop-outs did not differ in gender or disease complexity from participants that remained in the study. Information on how missing data was managed is explained in the data analysis section.

### Procedure

Data collection was undertaken by post when the participants were 16 years (T0) and 17 (T1) and 18 ½ years (T2). Eligible participants were sent a package containing information about

the study, an informed consent document, a set of questionnaires and pre-addressed return envelopes. To minimize non-response, reminders were sent after 2, 4 and 6 weeks [29].

## Measures

**Demographic characteristics** (age, sex, educational level) and **clinical data** (primary diagnosis) were collected from a background information questionnaire and the patients' medical records. Complexity of the heart disease was divided into simple, moderate or complex heart defects [30].

**Patient empowerment** was measured using the Gothenburg Young Persons Empowerment Scale-Congenital Heart Disease module (GYPES-CHD). This instrument comprises 15 items measured on a five-point Likert scale (strongly disagree to strongly agree). The scale measures five dimensions of patient empowerment: 1) knowledge and understanding; 2) personal control; 3) identity; 4) shared decision-making; and 5) enabling others. It is possible to calculate a subscale score for every dimension or a total score that ranges from 15–75, with higher scores denoting a higher level of empowerment [31]. The total score is used in the present study.

The following PROs to be assessed in relation to patient empowerment were chosen based on previous research that found significant effects between these variables and patient empowerment [21]. **Patient-reported health** was assessed with the generic and cardiac modules of the Pediatric Quality of Life Inventory 4.0 (PedsQL 4.0) [32]. The PedsQL 4.0 generic module comprises 23 items, measured on a five-point Liker scale (never to always). The scale covers the following dimensions: physical, social functioning, emotional and school functioning and it is possible to calculate subscale scores as well as a total patient-reported health score [32]. For the purpose of this study, the latter is included in the analyses.

**Communication skills** was measured with a subscale from the PedsQL 4.0 cardiac module [33]. The items are measured with the same Likert scale as the generic version. A higher score indicates fewer problems communicating with the healthcare providers and others about their CC [33].

QoL was measured with a linear analog scale (0–100), where a higher scoring indicates a better self-perceived QoL [34].

**Transition readiness** was assessed with two items from the Readiness for Transition Questionnaire (RTQ) [35]. These two items address readiness for taking over responsibility for their care and readiness for the transfer to adult care. The items are measured with a four-point Likert scale (not at all ready to completely ready) and the total score ranges from 2–8, with a higher score indicating the person is more ready for transition.

## Data analysis

Descriptive statistics were expressed in absolute numbers, percentages, means and standard deviations. To test mean differences between time points, one-way repeated measures ANOVAs were undertaken. Effect sizes were reported through partial eta squared values. Additionally, minimal clinically important differences (MCIDs) for all the scales were determined by a one standard error of measurement, which is a distribution-based method to define MCIDs [36].

To explore the longitudinal associations between patient empowerment and other PROs, random intercepts cross-lagged panel models (RI-CLPM) were used. This type of structural equation modelling is useful when analysing longitudinal, observational data to understand reciprocal relationships between variables [37,38]. RI-CLPM also allows to differentiate *between* and *within* person effects, that otherwise are not possible to be determine with a classic CLPM [39]. *Between* person effects focus on the differences in the variables of interest in

between persons (individuals who report a higher/lower level of empowerment also report a higher/lower level of the PRO compared to their peers), whereas the *within* person effects provide evidence on the relationship between variables (individuals who score a higher/lower level of empowerment than their expected score also tend to score higher/lower in their PRO score). Including a random intercept in the CLPM helps reduce biases in the estimates that are caused by uncontrolled between-person effects in variances and p-values, but not over fixed effects [39]. Additionally, by differentiating *between* and *within* person effects it is possible to identify who needs an intervention and modifiable targets, respectively. In the S1 File, a comparison between the model fit indices of the CLPM and the RI-CLPM is provided.

Fig 1 provides an example of the RI-CLPM used. This model follows the procedures described by Hamaker et al [39]. The figure shows that each patient empowerment score and the corresponding PRO score was decomposed into a stable between-person part and a within-person varying part. The between-person part is represented by the two random intercepts included in the figure (one for each construct). The model includes three types of relations: within-time relations, carry-over stability paths and cross-lagged relations. The latter are the ones of most interest in this study, they indicate how the variables influence each other, i.e. the extent to which changes from the PROs are predicted by patient empowerment and vice-versa [39]. The coefficient (β) from the cross-lagged relations indicates the extent to which a persons' change in deviation from their expected score in one PRO are predicted by deviations from their expected score on the other PRO from the previous measurement.

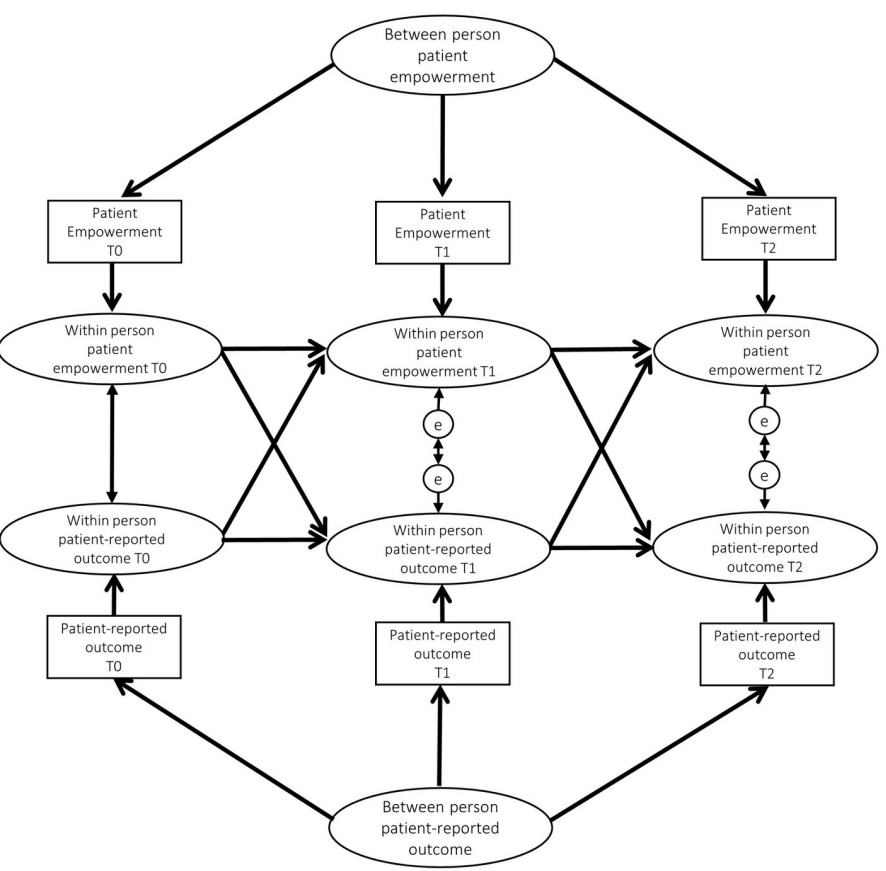

**Fig 1. Random intercepts cross-lagged model tested.**

To ensure that all variation was captured by the between and within person effects, all error variances of the observed scores were constrained to zero [40]. Given that there were no strong theoretical or empirical reasons to constrain the cross-lagged and carry-over relations, these are calculated fully free.

The aforementioned model was replicated four times, as we individually assessed the relationships between patient empowerment and transition readiness, patient-reported health, communication skills and QoL. For each of the four models, goodness-of-fit indices were reported: chi-square, comparative fit index (CFI), root mean square error of approximation (RMSEA) and standardised root mean square residual (SRMR). Models with a CFI >0.090 were considered to have an acceptable fit and good fit if it was >0.95. RMSEA and SRMR values of <0.08 were acceptable and <0.05 were a good fit [41]. Missing data was managed by the Full-Information Maximum Likelihood method (FIML), which is the recommended approach in structural equation modeling [42]. FIML produces estimates based on the assumption that missing data is at random, to confirm this assumption, Little's test for missing completely at random (MCAR) was undertaken. Results from this analysis indicated that the data was MCAR (p = 0.128).

Statistical analyses were performed using IBM SPSS Statistics for Windows version 27 and the Lavaan package in R. All tests were two-sided and the significance level was established at $p \leq 0.05$.

## Ethical approval and informed consent

Ethics approval to conduct the study was received from the Ethics board of Gothenburg, Sweden (no.931 15). According to Swedish regulations, persons between the ages of 15–18 years are able to provide assent in order to participate in research studies, independently on whether the parents give their approval [43]. In this project, in line with the aforementioned regulation, the young persons were the ones who provided assent and their parents were not required to provide consent on behalf of their child. All participants were asked to sign an informed consent document and were informed they could withdraw from the study at any time point.

## Results

### Demographic and clinical characteristics

The sample comprised of 59.4% (n = 82) boys. Overall, 13.8% (n = 19) had a mild CHD, 61.6% (n = 85) a moderate CHD and 24.6% (n = 34) a complex CHD.

### Temporal relationships

Longitudinal assessment of the study variables showed significant mean differences over the three time points (See Table 1).

Patient empowerment, patient-reported health, communication skills and transition readiness appeared to increase. On the other hand, QoL decreased between T0 and T1 and increased later on at T2. The changes between measurements were significant for QoL and transition readiness. Additionally, the effect size for the temporal changes for transition readiness was large. Altogether, between 3%-22% of the participants reported increases or decreases larger than the MCIDs between T0 and T2. Transition readiness was the PRO with the largest proportion of participants whose score increased.

Results from the four models tested are shown in Fig 2 and their corresponding model fit indexes are shown in Table 2. All fit indexes of the models had a good fit, with the exception for the one tested between QoL and patient empowerment. The latter had a RMSEA value above the expected range.

**Table 1. Longitudinal assessment of study variables.**

| | T0 Mean (±SD) | T1 Mean (±SD) | T2 Mean (±SD) | p value | Partial eta squared | MCIDs | Increase (%)* | Decrease (%)* |
|---|---|---|---|---|---|---|---|---|
| **Empowerment** | 52.63 (10.31) | 53.71 (11.81) | 54.96 (12.64) | 0.107 | 0.02 | 12.99 | 14.1 | 3.0 |
| **Patient-reported health** | 81.35 (14.57) | 81.62 (17.42) | 82.84 (12.87) | 0.500 | 0.00 | 17.51 | 9.0 | 5.0 |
| **Communication skills** | 78.90 (22.62) | 80.05 (20.57) | 83.60 (18.43) | 0.073 | 0.03 | 31.22 | 11.0 | 3.0 |
| **Quality of life** | 81.15 (20.47) | 77.06 (18.49) | 80.60 (15.16) | 0.044 | 0.04 | 25.31 | 10.8 | 16 |
| **Transition readiness** | 5.12 (1.64) | 5.59 (1.63) | 6.38 (1.47) | <0.001 | 0.23 | 2.38 | 22.4 | 3.0 |

MCIDs: Minimal clinically important difference.

*Proportion of patients that have an increase or decrease higher than the MCID between T0 and T2.

The between person associations (not shown in Fig 2) were only significant in the model between patient empowerment and communication skills, indicating that individuals with a higher level of empowerment across the time points reported better communication skills than individuals with lower patient empowerment (β = 0.46, p<0.01). On the within person level, there was a significant cross-lagged effect from transition readiness to patient empowerment between T1 and T2 (β = 0.39, p<0.05). This significant effect is interpreted as persons with deviations from their expected patient empowerment score at T2 were predicted by their level of transition readiness at T1 (Fig 2, panel B). None of the other models had significant cross-lagged effects (Fig 2, panels A-D).

There was a significant carry-over effect in QoL from T0 to T1 (β = 0.62, p<0.01), indicating that within person deviations from the expected QoL score at T0 predicted deviations from the expected QoL score at T1 (Fig 2, panel A). Additionally, a significant within time association was found between patient empowerment and transition readiness at T1 (β = 0.47, p<0.05). This can be translated as changes in the mean score of the patient empowerment at T1 were associated with changes in transition readiness at this same time point.

## Discussion

To the best of our knowledge, the present study is the only study that uses a longitudinal design and RI-CLPM to test the longitudinal associations between patient empowerment and other PROs. Through these analyses and the nature of the data, it is possible to determine which variable has a predictive effect over the other one and to separate between and within person effects, obtaining pure estimates from the relationships of these variables. Our results showed that in young persons with CHD, transition readiness predicted the score of patient empowerment when the participants were 18 ½ years, indicating the direction of effect.

Bravo and colleagues [8] provide one of the most thorough conceptual models on patient empowerment. In the model, they propose that taking an active role and having perceived control are indicators of patient empowerment, aspects that in our study are captured by transition readiness. Indeed, transition readiness in our study is defined as the "adolescents' readiness to assume complete responsibility for their care and their readiness to transfer to adult medical care" [35]. Unlike in Bravo and colleagues' conceptual model, our findings support the idea that transition readiness, rather than being an indicator of patient empowerment, it is actually a determinant of it.

The association between patient empowerment and transition readiness has been investigated before in a cross-sectional study, which found a significant association [44]. The current study adds to available evidence by providing details on the direction of effects between these two variables. Research that has focused on the association between empowerment and

**Panel A**

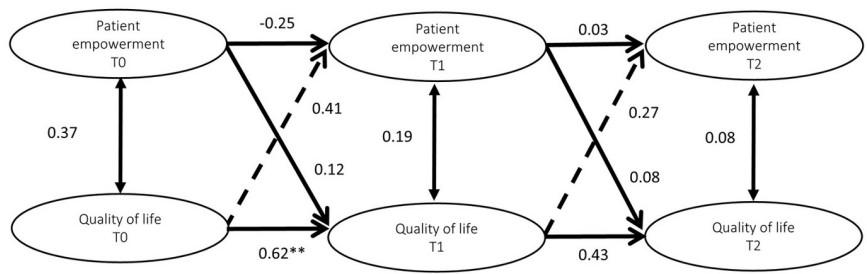

**Panel B**

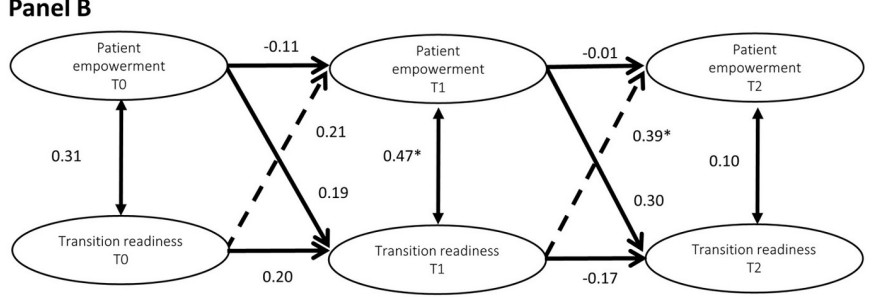

**Panel C**

**Panel D**

**Fig 2. Simplified results from the random intercept cross-lagged panel models.** This figure only shows the within person effects. * p≤0.05; ** p≤0.01; *** p≤0.001.

transition readiness is limited and research that focuses on variables associated with transition readiness are also mostly cross-sectional [45]. Transition readiness has previously been found to be significantly associated with several psychosocial factors such as self-efficacy, motivation and patient activation [45]. Therefore, it is reasonable for it to also be associated with patient empowerment.

**Table 2. Model fit statistics of the four random intercepts cross-lagged panel models tested.**

| Fit indexes | Quality of life | Transition readiness | Communication skills | Patient-reported health |
|---|---|---|---|---|
| **Comparative fit index (CFI)** | 0.981 | 1.000 | 0.999 | 0.998 |
| **Root mean square error of approximation (RMSEA)** | 0.180 | 0.000 | 0.044 | 0.061 |
| **Standardized root mean square residual (SRMR)** | 0.037 | 0.012 | 0.023 | 0.028 |
| Chi square test of model fit | 5.464 | 0.412 | 1.263 | 1.519 |
| Degrees of freedom | 1 | 1 | 1 | 1 |
| P value | 0.019 | 0.521 | 0.261 | 0.0218 |
| Normed chi$^2$ index ($x^2$/df) | 5.464 | 0.412 | 1.263 | 1.519 |

A CFI >0.090 was considered to have an acceptable fit and good fit if it was >0.95. RMSEA and SRMR values of <0.08 were acceptable and <0.05 were a good fit.

The predictive effect of transition readiness over patient empowerment can be potentially explained by the fact that young persons with CHD need to have the confidence or feel they are ready to assume the new role in front of them, before they can take a more active role in their health and lives and become empowered. A higher level of transition readiness can be associated with the adolescents having a certain level of awareness about their possibilities and what is required from them once they are transferred to adult care. This awareness is a prerequisite for choosing their own agenda (i.e. becoming empowered) [46].

Interestingly, the predictive effect of transition readiness was between T1 and T2, when the participants are 17 and 18 ½ years, respectively. Previous studies have found that transition readiness is associated with age and that it increases with age [45,47,48]. It is plausible that it is up to this point when their level of transition readiness is high enough to lead to an increment on patient empowerment. This fact is perhaps of relevance in understanding the longitudinal effects of transition readiness and the potential point when improvements over patient empowerment and perhaps other outcomes can be expected and therefore, measured. Future studies could evaluate the association between age, transition and patient empowerment, more specifically determining whether age is a mediating factor between these two variables.

Even when there is literature suggesting patient empowerment is associated with other variables in the long-term [8,49], there is still a lack of longitudinal research [50], meaning the available evidence supporting current models on patient empowerment is relatively low. This study is one of the first ones in providing longitudinal evidence that informs the potential predictive value of patient empowerment. Our findings are an initial step in more thoroughly understanding the theoretical framework of patient empowerment and how this construct associates with other variables.

The lack of significant cross-lagged effects with the other evaluated variables, such as QoL, might be associated with the fact that there could be a mediating variable that is not accounted for in the models. It is plausible that the effect of patient empowerment is mediated by a third variable. According to Palumbo [51], the variables that mediate the relationship between patient empowerment and other PROs have been overlooked. Palumbo [51] suggests that self-efficacy, self-management, health literacy and even patient activation could be some of the variables that strengthen the association between patient empowerment and other outcomes.

Another possible explanation for why we did not find significant cross-lagged effects could be that the level of patient empowerment of the participants was not sufficiently high to be associated with improvements of other PROs later on. There is no available information on what constitutes a high level of patient empowerment or even the necessary amount of patient empowerment to trigger improvements in other variables. There is evidence that empowering interventions have led to improvement in different outcomes, such as QoL [23,24,52]. Perhaps

in the case of our participants, they did not have a sufficiently high level of patient empowerment or even a sufficiently high level of the other PROs to achieve this.

It is also worth noting that cross-lagged analyses are dependent on the length of the time interval between measurements, meaning that other researchers who study the same models can potentially obtain different estimates, merely due to the time interval they use [53]. This particular aspect could also explain why we only found a significant cross-lagged effect and why perhaps other researchers in the future may find opposite results.

Our findings are the first from a longitudinal study that provide evidence on the associations between patient empowerment and other PROs across time. However, more longitudinal research is necessary to understand the predictive effect (or lack thereof) of patient empowerment. By doing so, current theoretical and conceptual models can be revised and interventions aiming to improve patient empowerment can be better designed.

## Methodological issues

This study has several strengths: 1) longitudinal data on patient empowerment are scarce and therefore not many studies have been capable of assessing longitudinal associations in relation to this construct; 2) all the questionnaires used had been previously validated in young persons with chronic conditions; 3) data collections were done within the same time frame for all participants; and 4) the study involved data from 7 different hospitals, which increases the generalizability of the results.

However, some limitations ought to be considered when interpreting the results. First, we have a small sample size. While literature on RI-CLPM and structural equation modeling has not come to an agreement on the required sample in order to convey proper models, a minimum of 200 participants has been suggested as the threshold [41]. Second, considering the sample size, it was not possible to test models that are more complex and control the effect of different variables. Third, our sample only included young persons who were between 16–18 ½ years old, which means these results might not provide evidence beyond this period in adolescence. Fourth, our study includes three different time points, which is the minimum for RI-CLPM, and this increases the risk that our study is underpowered. Fifth, dropouts are always a possibility when undertaking several data collections. We started with 138 patients at T0, and unfortunately, had only 101 by T1, despite our best efforts to promote participation.

## Conclusions

Our study is the first longitudinal study to attempt to describe the longitudinal associations between patient empowerment and other PROs, as well as the potential predictive value of empowerment. The present study provides evidence on the predictive value of transition readiness over patient empowerment. This means increments in transition readiness are associated with positive changes over patient empowerment. Researchers or clinicians working in transitional care can reflect on whether patient empowerment should be an outcome to be measured, in light of the predictive effect that transition readiness has over this outcome.

## Supporting information

**S1 File. Model comparison of the CLPM and RI-CLPM models.**
(DOCX)

## Acknowledgments

We would like to thank Koen Raymaekers for his support and comments throughout the data analysis and revision of the manuscript.

## Author Contributions

**Conceptualization:** Mariela Acuña Mora, Carina Sparud-Lundin, Philip Moons, Ewa-Lena Bratt.

**Data curation:** Mariela Acuña Mora.

**Formal analysis:** Mariela Acuña Mora.

**Funding acquisition:** Philip Moons, Ewa-Lena Bratt.

**Investigation:** Mariela Acuña Mora, Carina Sparud-Lundin, Philip Moons, Ewa-Lena Bratt.

**Methodology:** Mariela Acuña Mora, Carina Sparud-Lundin, Philip Moons, Ewa-Lena Bratt.

**Project administration:** Mariela Acuña Mora, Carina Sparud-Lundin, Eva Fernlund, Shalan Fadl, Kazamia Kalliopi, Annika Rydberg, Åsa Burström, Katarina Hanseus, Philip Moons, Ewa-Lena Bratt.

**Resources:** Mariela Acuña Mora, Eva Fernlund, Shalan Fadl, Kazamia Kalliopi, Annika Rydberg, Åsa Burström, Katarina Hanseus.

**Software:** Mariela Acuña Mora.

**Supervision:** Carina Sparud-Lundin, Philip Moons, Ewa-Lena Bratt.

**Visualization:** Mariela Acuña Mora.

**Writing – original draft:** Mariela Acuña Mora.

**Writing – review & editing:** Carina Sparud-Lundin, Eva Fernlund, Shalan Fadl, Kazamia Kalliopi, Annika Rydberg, Åsa Burström, Katarina Hanseus, Philip Moons, Ewa-Lena Bratt.

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
