## [Decision Letter · Decision Letter 0]

17 Mar 2022

PONE-D-21-31558The longitudinal association between patient empowerment and patient-reported outcomes: what is the direction of effect?PLOS ONE

Dear Dr. Acuña Mora,

Thank you for submitting your manuscript to PLOS ONE. After careful consideration, we feel that it has merit but does not fully meet PLOS ONE’s publication criteria as it currently stands. Therefore, we invite you to submit a revised version of the manuscript that addresses the points raised during the review process.

Please address the queries raised by the reviewers on the analysis methods, and the need for further clarity and detail in the objectives and some of the methods. 

We look forward to receiving your revised manuscript.

Kind regards,

Kathleen Finlayson

Academic Editor

PLOS ONE

Journal Requirements:

2. You indicated that you had ethical approval for your study. In your Methods section, please ensure you have also stated whether the research ethics committee or IRB specifically waived the need for their consent.

Reviewers' comments:

Reviewer's Responses to Questions

**Comments to the Author**

1. Is the manuscript technically sound, and do the data support the conclusions?

Reviewer #1: Partly

Reviewer #2: Yes

2. Has the statistical analysis been performed appropriately and rigorously? 

Reviewer #1: No

Reviewer #2: No

3. Have the authors made all data underlying the findings in their manuscript fully available?

Reviewer #1: No

Reviewer #2: Yes

4. Is the manuscript presented in an intelligible fashion and written in standard English?

Reviewer #1: Yes

Reviewer #2: Yes

5. Review Comments to the Author

Reviewer #1: Page 7, line 111: If you imputed observations for the dropout observations you should explicitly state that

Page 8, line 132: When empowerment (or continuous PROs) is treated as an outcome, is there a validated threshold for the average difference in score beyond which the results are considered as important? If yes, please use it to interpret the findings

Page 8, line 143: Similar to the previous comment, is there a threshold for the difference in PedsQL 4.0 should regard as important? Is it dependent on baseline values?

Page 9, line 154: You need to evaluate skewness for the distribution of your continuous measurements, if skewed you should report your descriptives in terms median and interquartile range

Page 9, line 156: Large partial eta squared valued may reflect small sampling variability rather effectiveness of empowerment. To demonstrate effectiveness please report mean differences or other similar measures

Page 9, line 167: For continuous outcomes, random effects (including random intercept) accounts for bias in variances and p-values but not biases in fixed effects. Please re-word your sentence to make it clear that you are not referring to fixed effects

Page 10, line 193: By using FIML, you are assuming that the missing observations are missing at random. Please justify how you landed to this assumption

Page 11, line 214: You have 3 time points so the comparison shouldn’t be confined to only 2 points (I’m aware that Table 1 has all 3 time points)

Table 1, Quality of Life: Because of lack of linear trend, I don't find the reported P value and partial eta squared for QoL to be useful in this instance (QoL decreased at T2 but increased at T3). Also, it is a little puzzling that SE decreases over time although there are dropouts in subsequent time points. Even if you imputed, I would expect FIML to account for uncertainty due imputed data

Page 12, line 218: QoL did not decrease consistently. It decreased at T1 and increased from T1 to T2

Pages 13 & 14: How do we interpret beta? Is it standardized mean difference (standardized to have mean 0 and variance 1)? If so, it is hard to tell if large values for beta imply a large difference in means or small variance. It is informative to report mean differences in their natural form along with 95% CI.

Page 13, line 233: The meaning of “effect” seems to differ across the manuscript. Please make sure that you are consistent with the usage of the term “effect”. Also, what you describe as “effect” is simply a measure of compatibility of your model with the null hypothesis

Page 14, line 245: The asterisks in p values seems to suggest the smaller is the p values the larger is the magnitude of effectiveness. Please note, small p-values may merely reflect small SEs, as such, they shouldn't be treated as measures of strength of effectiveness

Page 18, lines 342-349: There is a difference between predicting and evaluating whether empowerment affects PRO and vice versa. Your conclusion suggests your interest was in predicting whereas the research question suggest the interest is in estimating the effects. More clarity is needed on your objectives

Reviewer #2: This manuscript considers secondary analysis of data generated from a randomized clinical trial. The objective here is to examine the direction of effects between patient empowerment and PROs in young subjects with CHD. The design generated a longitudinal study, and the analysis here considers comparing 2 groups. I have the following queries, which, when addressed, will strengthen the analysis.

1. It would be great to provide some ballpark sample size/power estimate, wrt. the desired effect size in mind. This maybe important to replicate this design. For example, 2 groups, each with 70 subjects, were considered. So, what is the resultant power wrt. this sample size, considering the longitudinal design, and the primary response variable (the composite score).

2. A one-way repeated measures ANOVA was used to analyze the longitudinal outcomes. ANOVAs depend heavily on Gaussianity assumptions of the response variable. How was that assessed? If failed, alternative methods are needed, such as Friedman's test, maybe needed.

3. Same goes for the RI-CLPM modeling, which assumes Gaussianity . Further, to better promote the RI-CLPM, some comparisons are needed with the basic CLPM. Authors may follow this link if they want to:

https://johnflournoy.science/2017/10/20/riclpm-lavaan-demo/

6. PLOS authors have the option to publish the peer review history of their article (what does this mean?). If published, this will include your full peer review and any attached files.

Reviewer #1: No

Reviewer #2: No

---

## [Author Response · Author response to Decision Letter 0]

24 May 2022

Reviewer #1

Page 7, line 111: If you imputed observations for the dropout observations you should explicitly state that

Information on how missing data was managed is provided under the “Data analysis” section. For the RI-CLPM missing data was managed through FIML. However, data was not imputed for the one-way repeated measures ANOVA. 

Page 8, line 132: When empowerment (or continuous PROs) is treated as an outcome, is there a validated threshold for the average difference in score beyond which the results are considered as important? If yes, please use it to interpret the findings

Minimal clinically important differences (MCIDs) were calculated for all scales using a distribution-based method, which was established as one standard error of measurement [1]. This approach is relatively stable across populations and considers the precision of the measure [1, 2]. This information has now been included in the “Data analysis” section, as well as in Table 1. Furthermore, the interpretation of this information has been included in the “Results” (See page 10, lines 173-175).

Page 8, line 143: Similar to the previous comment, is there a threshold for the difference in PedsQL 4.0 should regard as important? Is it dependent on baseline values?

As mentioned previously, MCIDs were determined based on a one standard error of measurement. 

Page 9, line 154: You need to evaluate skewness for the distribution of your continuous measurements, if skewed you should report your descriptives in terms median and interquartile range

Normality of the data was assessed through histograms and Q-Q plots as well as with the Kolmogorov Smirnov och Shapiro Wilk tests. To counter the potential bias of these approaches, normality was also assessed through the skewness and kurtosis of the data. Acceptable skewness was between -2/+2 and kurtosis between -7/+7 [3]. An assessment of all this data indicated the data was approximately normally distributed. Therefore, the report of means and standard deviations is fitting. 

Page 9, line 156: Large partial eta squared valued may reflect small sampling variability rather effectiveness of empowerment. To demonstrate effectiveness please report mean differences or other similar measures

The partial eta square for patient empowerment was 0.02, which is a small effect size [4]. The only outcome that had a large effect size was transition readiness (partial eta square= 0.23). To account for the potential bias of this estimate, we are also reporting MCIDs in Table 1. 

Page 9, line 167: For continuous outcomes, random effects (including random intercept) accounts for bias in variances and p-values but not biases in fixed effects. Please re-word your sentence to make it clear that you are not referring to fixed effects

This sentence has been rephrased to highlight the fact that the random intercept does not account for biases in fixed effects (See page 11, lines 187-188). 

Page 10, line 193: By using FIML, you are assuming that the missing observations are missing at random. Please justify how you landed to this assumption 

Indeed, FIML assumes that data are missing at random. In order to confirm this assumption, Little’s missing completely at random (MCAR) test was undertaken [5, 6]. This analysis was non-significant (p=0.128), confirming the data are missing at random. This information has been included in the “data analysis” section. 

Page 11, line 214: You have 3 time points so the comparison shouldn’t be confined to only 2 points (I’m aware that Table 1 has all 3 time points)

This was a typographical error that has been corrected. Indeed, the comparison was done for the three time points. 

Table 1, Quality of Life: Because of lack of linear trend, I don't find the reported P value and partial eta squared for QoL to be useful in this instance (QoL decreased at T2 but increased at T3). Also, it is a little puzzling that SE decreases over time although there are dropouts in subsequent time points. Even if you imputed, I would expect FIML to account for uncertainty due imputed data

The information reported in table 1 is not the standard error (SE), but rather the standard deviation (SD). Unfortunately, this was a typographical error that was not identified before the submission of the manuscript. This error indeed influences the interpretation of the table and expected changes as the sample size decreases. We have corrected the typographical error in Table 1. It is worth noting, that FIML was only used for the RI-CLPM and not for the one-way repeated measures ANOVAs. 

Page 12, line 218: QoL did not decrease consistently. It decreased at T1 and increased from T1 to T2. 

The reviewer is correct, and this sentence has been revised to clarify that quality of life decreased initially and later increased again. 

Pages 13 & 14: How do we interpret beta? Is it standardized mean difference (standardized to have mean 0 and variance 1)? If so, it is hard to tell if large values for beta imply a large difference in means or small variance. It is informative to report mean differences in their natural form along with 95% CI.

While it is possible to report 95% CI, it is not standard practice to report this information. The β coefficient from the cross-lagged relations indicates the extent to which for instance a persons’ change in deviation from their expected quality of life score is predicted by deviations from their expected score on patient empowerment from the previous measurement. The reported coefficients in the manuscript are standardized. 

Page 13, line 233: The meaning of “effect” seems to differ across the manuscript. Please make sure that you are consistent with the usage of the term “effect”. Also, what you describe as “effect” is simply a measure of compatibility of your model with the null hypothesis

Across the manuscript when we write “effect”, we refer to the influence one variable has over the other. This effect can be for instance the influence patient empowerment has over another PRO in a follow-up measurement or the influence patient empowerment has over itself. 

We have revised the manuscript to confirm the use of the word “effect” is consistent throughout the manuscript. 

Page 14, line 245: The asterisks in p values seems to suggest the smaller is the p values the larger is the magnitude of effectiveness. Please note, small p-values may merely reflect small SEs, as such, they shouldn't be treated as measures of strength of effectiveness. 

As the reviewer indicates, p-values are not indicators of the effect’s magnitude. P-values are indicators of how likely it is to have found a particular set of results if the null hypothesis is true. If the researcher wants to indicate the magnitude of a significant result, then other measures such as Cohen’s D or eta square are more relevant, since these measures will indicate whether the effect is small, medium or large. Nonetheless, our reporting of the p-values as such is common practice. The more asterisks (*), the smaller the p value, and this should be interpreted as the probabilities of having found these results if the null hypothesis is true, are smaller. 

Page 18, lines 342-349: There is a difference between predicting and evaluating whether empowerment affects PRO and vice versa. Your conclusion suggests your interest was in predicting whereas the research question suggest the interest is in estimating the effects. More clarity is needed on your objectives

The aim of our study was to examine the longitudinal associations between patient empowerment and other patient-reported outcomes. This examination entailed both, estimating the effects and determining which variable has a predicting effect over the other. Therefore, the information provided in the conclusions is in line with this. We decided to focus on the cross-lagged relations (i.e., predictive effect), because it’s the most relevant finding from the study.

Reviewer #2

This manuscript considers secondary analysis of data generated from a randomized clinical trial. The objective here is to examine the direction of effects between patient empowerment and PROs in young subjects with CHD. The design generated a longitudinal study, and the analysis here considers comparing 2 groups. I have the following queries, which, when addressed, will strengthen the analysis.

It would be great to provide some ballpark sample size/power estimate, wrt. the desired effect size in mind. This maybe important to replicate this design. For example, 2 groups, each with 70 subjects, were considered. So, what is the resultant power wrt. this sample size, considering the longitudinal design, and the primary response variable (the composite score).

The sample size for this study is based on the sample size calculation that was made for a larger study. In the latter, the calculation was based on an improvement of patient empowerment of 5.25 points, an α=0.05 and power=80%. This meant we needed 63 patients in each arm. However, to compensate for drop-outs, 70 patients were included in each arm. This information has been included in the “Study population” section. 

A one-way repeated measures ANOVA was used to analyze the longitudinal outcomes. ANOVAs depend heavily on Gaussianity assumptions of the response variable. How was that assessed? If failed, alternative methods are needed, such as Friedman's test, maybe needed.

One of the assumptions for one-way repeated measures ANOVAs is that the residuals should be normally distributed. The residuals represent the difference between each individual observation and the group’s mean from where the observation came from. Hence, the raw data does not need to be normally distributed. The normal distribution of the residuals was evaluated through their skewness and kurtosis. These parameters were selected given that tests such as Shapiro-Wilk and Kolmogorov Smirnov tests are sensitive to sample sizes and histograms, P-P plots and Q-Q plots, while useful, rely heavily on the interpretation of the observer. Cut-off values of -2/+2 for skewness and between -7/+7 for kurtosis were used to determine whether the residuals were normally distributed [3]. While these cut-off values might seem too broad, ANOVAs are analyses that also tolerate certain deviations from the normal distribution.

Same goes for the RI-CLPM modeling, which assumes Gaussianity. Further, to better promote the RI-CLPM, some comparisons are needed with the basic CLPM. Authors may follow this link if they want to:

https://johnflournoy.science/2017/10/20/riclpm-lavaan-demo/

RI-CLPM models assume multivariate normality for continuous outcomes. However, testing for multivariate normality its difficult to assess, given that small deviations from normality are easily detected in large samples and in small samples, power might not be enough to detect them [7]. Therefore, it is often the case that the inspection of each variable (univariate normality) through their skewness and kurtosis is a useful tool [3]. This is important because multivariate normality is not possible without univariate normality. As it was mentioned in a previous response, the variables’ distribution was assessed, and their skewness and kurtosis were within the accepted ranges. However, since there was the risk that our data was not complying with the multivariate normality assumption, as an estimator we used “robust maximum likelihood (MLR)”. This estimator is meant for variables with non-normal distributions [7].

The CLPM is nested in the RI-CLPM, so comparisons of these models are in occasions presented in the literature. We have decided to provide the results from these comparisons as a Supplementary File. This was done so because the paper focuses on evaluating the longitudinal relationships between patient empowerment and other PROs, rather than the comparison across different structural equation models. We refer to the Supplementary File in page 11, lines 189-190. 

References 

1. Sedaghat AR. Understanding the Minimal Clinically Important Difference (MCID) of Patient-Reported Outcome Measures. Otolaryngology–Head and Neck Surgery. 2019;161(4):551-60.

2. Crosby RD, Kolotkin RL, Williams GR. Defining clinically meaningful change in health-related quality of life. Journal of Clinical Epidemiology. 2003;56(5):395-407.

3. Byrne BM. Structural equation modeling with AMOS : basic concepts, applications, and programming. Mahwah, N.J.: Mahwah, N.J. : Lawrence Erlbaum Associates; 2001.

4. Lakens D. Calculating and reporting effect sizes to facilitate cumulative science: a practical primer for t-tests and ANOVAs. 2013;4(863).

5. Roderick JAL. A Test of Missing Completely at Random for Multivariate Data with Missing Values. Journal of the American Statistical Association. 1988;83(404):1198-202.

6. Li C. Little's Test of Missing Completely at Random. The Stata Journal. 2013;13(4):795-809.

7. Kline Rex. Principles and practices of strucutal equation modeling. Fourth edition ed. United States: Guilford Publications; 2016.

---

## [Decision Letter · Decision Letter 1]

25 Jul 2022

PONE-D-21-31558R1The longitudinal association between patient empowerment and patient-reported outcomes: what is the direction of effect?PLOS ONE

Dear Dr. Acuña Mora,

Thank you for submitting your manuscript to PLOS ONE. After careful consideration, we feel that it has merit but does not fully meet PLOS ONE’s publication criteria as it currently stands. Therefore, we invite you to submit a revised version of the manuscript that addresses the points raised during the review process.

The manuscript has been evaluated by two reviewers, and their comments are available below.

The reviewers note concerns about the statistical analyses presented and request re-analyses be completed.

Could you please carefully revise the manuscript to address all comments raised?

We look forward to receiving your revised manuscript.

Kind regards,

Thomas Phillips, PhD

Staff Editor

PLOS ONE

Journal Requirements:

Reviewers' comments:

Reviewer's Responses to Questions

**Comments to the Author**

1. If the authors have adequately addressed your comments raised in a previous round of review and you feel that this manuscript is now acceptable for publication, you may indicate that here to bypass the “Comments to the Author” section, enter your conflict of interest statement in the “Confidential to Editor” section, and submit your "Accept" recommendation.

Reviewer #1: (No Response)

Reviewer #2: All comments have been addressed

2. Is the manuscript technically sound, and do the data support the conclusions?

Reviewer #1: Yes

Reviewer #2: (No Response)

3. Has the statistical analysis been performed appropriately and rigorously? 

Reviewer #1: Yes

Reviewer #2: (No Response)

4. Have the authors made all data underlying the findings in their manuscript fully available?

Reviewer #1: Yes

Reviewer #2: (No Response)

5. Is the manuscript presented in an intelligible fashion and written in standard English?

Reviewer #1: Yes

Reviewer #2: (No Response)

6. Review Comments to the Author

Reviewer #1: The authors fully addressed most of the comments. There is one issue on MCID (Page 10, line 172) that require more clarification from the authors. Specifically, they use the standard error (SE) less than 1 to reflect whether the scale is showing MCID. Since SE are sample size dependent, their values might not reflect clinical importance of the scale measurement results. For example, a very large study done in a population with very low QoL may yield a SE less than 1, and using SE one will erroneously conclude that the QoL in this population is high. In fact, SE in such a population is guaranteed to fall below 1 as long as you keep on increasing the sample size. On the other hand, a modest size study done on a population with high QoL (on average) may result in SE greater than 1. Perhaps you should consider using SD instead, as it reflects variability between subject measurements (on average) rather than the variability on a sample statistic.

Reviewer #2: (No Response)

7. PLOS authors have the option to publish the peer review history of their article (what does this mean?). If published, this will include your full peer review and any attached files.

Reviewer #1: No

Reviewer #2: No

---

## [Author Response · Author response to Decision Letter 1]

18 Aug 2022

Dr. Thomas Philips (Staff Editor)

PLOS ONE 

Submission of manuscript

Dear Dr. Philips, 

Please find attached the revised version of the manuscript entitled “The longitudinal association between patient empowerment and patient-reported outcomes: what is the direction of effect?”. We would like to resubmit this manuscript for consideration by PLOS ONE.

We thank you for the opportunity to revise and resubmit the manuscript. Furthermore, we would like to thank the reviewer for the constructive comment. Enclosed is a letter summarizing the rebuttal on the comment from the reviewer. Corrections and additions made in the manuscript have been marked using the “track changes” option of MS Word. 

We hope that we have sufficiently addressed your comments. Thank you for considering this manuscript for publication in the PLOS ONE. 

On behalf of all co-authors,

Mariela Acuña Mora.

Reviewer #1

The authors fully addressed most of the comments. There is one issue on MCID (Page 10, line 172) that require more clarification from the authors. Specifically, they use the standard error (SE) less than 1 to reflect whether the scale is showing MCID. Since SE are sample size dependent, their values might not reflect clinical importance of the scale measurement results. For example, a very large study done in a population with very low QoL may yield a SE less than 1, and using SE one will erroneously conclude that the QoL in this population is high. In fact, SE in such a population is guaranteed to fall below 1 as long as you keep on increasing the sample size. On the other hand, a modest size study done on a population with high QoL (on average) may result in SE greater than 1. Perhaps you should consider using SD instead, as it reflects variability between subject measurements (on average) rather than the variability on a sample statistic.

As the reviewer points out, the standard error (SE) is sample size dependent. This means that as the sample size increases, the smaller the SE becomes and vice versa. This characteristic of the SE can have an impact on the way the MCID is calculated and hence, interpreted. While other distribution-based methods to calculate the MCID are available, for instance methods that use the standard deviation (SD), there are also limitations when following this approach. One of the most relevant limitations of using the SD is that the MCID calculation may be sample dependent, and thus the calculation is less generalizable [1, 2]. MCIDs that rely on the SE allow for a higher generalizability of the scores [1, 2]. Another benefit of using the SEM is that it is expressed in the original metric of a measure, which facilitates the interpretation of the results [2].

Despite its limitations, the calculation of the MCID using the SE is frequently used in studies [2, 3, 4] and the available evidence suggests that this approach is reliable for identifying MCID [2]. It is also worth noting that the MCID varies based on the sample and the context. Therefore, it should not be accepted as a universal fact [1]. The interpretation of the MCID should be made with caution, while keeping in mind the limitations of the selected approach. One solution could be to use different approaches when reporting the MCID so the reader can make a comparison of the values based on different methods [1, 5]. We have reflected on this option but decided not do so, because it would bring the discussion on the MCID beyond perspective, and it would distract the reader from the main focus of the article, being the direction of effects. Thank you for your understanding. 

References 

1. Sedaghat AR. Understanding the Minimal Clinically Important Difference (MCID) of Patient-Reported Outcome Measures. Otolaryngology–Head and Neck Surgery. 2019;161(4):551-60.

2. Wyrwich, K.W., Tierney, W.M. & Wolinsky, F.D. Further Evidence Supporting an SEM-Based Criterion for Identifying Meaningful Intra-Individual Changes in Health-Related Quality of Life. J Clin Epidemiol. 1999; 52(9), pp.861–873. 

3. Ousmen, A., Touraine, C., Deliu, N. et al. Distribution- and anchor-based methods to determine the minimally important difference on patient-reported outcome questionnaires in oncology: a structured review. Health Qual Life Outcomes. 2018; 16(228). https://doi.org/10.1186/s12955-018-1055-z

4. Mouelhi, Y. et al. How is the minimal clinically important difference established in health-related quality of life instruments? Review of anchors and methods. Health Qual Life Outcomes. 2020; 18(1), p.136.

5. Crosby RD, Kolotkin RL, Williams GR. Defining clinically meaningful change in health-related quality of life. J Clin Epidemiol. 2003;56(5):395-407.

---

## [Editor Report · Decision Letter 2]

25 Oct 2022

The longitudinal association between patient empowerment and patient-reported outcomes: what is the direction of effect?

PONE-D-21-31558R2

Dear Dr. Acuña Mora,

We’re pleased to inform you that your manuscript has been judged scientifically suitable for publication and will be formally accepted for publication once it meets all outstanding technical requirements.

Kind regards,

James Mockridge

Staff Editor

PLOS ONE
---

## [Editor Report · Acceptance letter]

2 Nov 2022

PONE-D-21-31558R2 

The longitudinal association between patient empowerment and patient-reported outcomes: what is the direction of effect? 

Dear Dr. Acuña Mora:

I'm pleased to inform you that your manuscript has been deemed suitable for publication in PLOS ONE. Congratulations! Your manuscript is now with our production department. 

Kind regards, 

on behalf of

Dr James Mockridge 

Staff Editor

PLOS ONE